# Pre-Slaughter Stunning Methods Influence the Meat Quality of *Arapaima gigas* Fillets

**DOI:** 10.3390/ani14081155

**Published:** 2024-04-11

**Authors:** Jucilene Braitenbach Cavali, Sheyla Cristina Vargas Baldi, Ana Sabrina Coutinho Marques Rocha, Erica Eloy da Silva, Carla Taveira Nunes, Emerson Carlos Soares, Sandro de Vargas Schons, Renato Zanella, Rute Bianchini Pontuschka, Jerônimo Vieira Dantas Filho

**Affiliations:** 1Laboratório de Análises Físico-Químicas e Microbiológicas, Universidade Federal de Rondônia, Presidente Médici 76916-000, Brazil; jcavali@unir.br (J.B.C.); anasabrina.coutinho@gmail.com (A.S.C.M.R.); eecuevaseloy@gmail.com (E.E.d.S.); nutricao@facsaopaulo.edu.br (C.T.N.); rutepont@unir.br (R.B.P.); 2Programa de Pós-Graduação em Sanidade e Produção Animal Sustentável na Amazônia Ocidental, Universidade Federal do Acre, Rio Branco 69920-900, Brazil; 3Aquaeficiência Consultoria e Gestão em Aquacultura, São Paulo 05508-220, Brazil; sheyla@consulting-fs.com.br; 4Departamento de Medicina Veterinária e Agronegócio, Centro Universitário São Lucas, UniSL JPR AFYA, Ji-Paraná 76907-524, Brazil; 5Programa de Pós-Graduação em Aquicultura, Universidade Estadual Paulista “Julio de Mesquita Filho”, Jaboticabal 76907-524, Brazil; 6Programa de Pós-Graduação em Ciências Ambientais, Universidade Federal de Rondônia, Rolim de Moura 76940-000, Brazil; sandroschons@unir.br; 7Centro de Aquicultura e Ecologia Aquática, Universidade Federal de Alagoas, Rio Largo 57100-000, Brazil; soaemerson@gmail.com; 8Programa de Pós-Graduação em Química, Universidade Federal de Santa Maria, Santa Maria 97105-900, Brazil; renato.zanella@ufsm.br

**Keywords:** animal welfare, aquaculture, electronarcosis, fish technology

## Abstract

**Simple Summary:**

It was confirmed that electronarcosis (EE) is a better method of pre-slaughter stunning for the meat quality of *Arapaima gigas* fillets from fish farming; the other methods are ice asphyxiation (IA), air asphyxiation (AA) and hypothermia followed by bleeding (HB), which showed fewer interesting results. Data were obtained from a pH analysis, rigor mortis index (RI), water activity (Aw), instrumental texture, blood glucose and instrumental colourimetry. During the study, for up to 15 days of refrigerated storage, the methods provided pH values below 6.0. EE resulted in better texture assignments in the fillets. The blood glucose values indicate that the fish subjected to EE were less stressed. Regarding instrumental colourimetry, the fillets submitted to EE and HB showed greater luminosity, those subjected to AA exhibited greater red-green colour intensity and those subjected to EE showed greater yellow-blue colour intensity. Therefore, the fish did not suffer stress with electronarcosis, and the fillets showed better preservation, juiciness and tenderness.

**Abstract:**

The aim of this study was to evaluate the influence of different stunning methods on the meat quality of *Arapaima gigas* fillets from fish farming. A total of 48 specimens of *A. gigas* in the weight class 11.1 to 14.0 kg were investigated; these fish were subjected to different stunning methods for slaughter: ice asphyxia (IA), air asphyxia (AA), electronarcosis (EE) and hypothermia followed by bleeding (HB). Then, data were obtained from the analysis of pH, rigor mortis index (RI), water activity (Aw), instrumental texture (compression strength, firmness and adhesiveness) and blood glucose and via instrumental colourimetry. During the study, for up to 15 days of refrigerated storage, the methods provided pH values below 6.0. *A. gigas* submitted to EE and HB remained longer in the pre-rigor status. In addition, they expressed lower percentages of Aw. The EE method resulted in better texture assignments in the fillets. The blood glucose values indicated that the fish subjected to EE were less stressed. Concerning instrumental colourimetry, the fillets submitted to EE and HB showed greater luminosity; the fillets subjected to AA showed greater red-green colour intensity, while the fillets subjected to EE showed greater yellow-blue colour intensity. Therefore, the fish did not suffer stress with electronarcosis, and the fillets showed better preservation, juiciness, and tenderness.

## 1. Introduction

It is known that the humane treatment of production animals from birth to slaughter provides benefits, such as lower rates of injuries, stress and mortality. Consequently, higher-quality meat is obtained with a gain in the assigned value. Stress can relate to several factors such as feeding, a high stocking density, a high parasite index, handling, capture and slaughter [1]. Slaughter, which was considered a low-tech operation, became a cause for concern when there was an understanding that animal management from rural property until slaughter influences the meat quality [2]. Generally, the stunning slaughter methods used by the fish industry are considered stressful. However, stunning methods and slaughter via cranial percussion, cutting the spinal cord followed by cutting the gills, an overdose of anaesthetics and electronarcosis can be considered non-stressful methods because they cause rapid stunning and less suffering to the fish [3].

Slaughter can be subdivided into two stages: the first is when the fish is stunned (pre-slaughter stunning), and the second is when it is slaughtered (the death). Pre-slaughter stunning can occur in a variety of ways, such as hypothermia, electronarcosis, anaesthetic overdose, head percussion, or asphyxiation. The slaughter of a fish can be carried out by cutting the gills, bleeding, eviscerating, decapitating and cranial percussion. Fish can be subjected to both phases or directly to the second phase. Electronarcosis is one of the most studied pre-slaughter stunning methods in several fish species [4]. It consists of passing an electric current through the water or directly through the fish until a complete loss of consciousness [3]. This method is considered non-stressful because it is quick and causes less suffering. Several factors can influence the effectiveness of electronarcosis, such as the type of electrical current (alternating current (AC) or direct current (DC)), voltage, frequency (Hertz) [5,6], the duration of the electric current (seconds) [7] and water conductivity [6]. In sea bass (*Dicentrarchus labrax*) subjected to electrical stunning (from 40 to 120 V and from 50 to 400 Hz) for 10 s, no recovery of consciousness was observed after 20 min [8]. In tench (*Tinca tinca*) subjected to electrical stunning (51 V) for 30 s, the onset of rigor mortis was faster when compared to the application of head percussion, live cooling, or CO_2_ narcosis, although without compromising other quality parameters, such as pH and meat colour and texture [4]. Therefore, studies on various factors of humane methods of slaughter, such as electric shock, are of great relevance.

*Arapaima gigas* (Schinz, 1822) is an endemic fish from the Amazon Basin that mainly inhabits floodplain lakes and flooded forests. This fish is a traditional Amazonian cuisine and a valuable fishing resource [9]. However, currently, environmental authorities, such as the agency Instituto Chico Mendes de Conservação da Biodiversidade (ICMBio), consider the natural populations of *A. gigas* sensitive to extinction. Others believe it an invasive species in some watersheds, and such facts have generated impediments to licensing and embargoed the development of its production chain in the Western Amazon. Therefore, it currently corresponds to less than 5% of the 57.2 thousand tons of native fish produced in the state of Rondônia [10].

Despite the commercial importance of *A. gigas*, studies on pre-slaughter stunning methods, fish behaviour and the meat quality of native species are still few in the literature [11]. Therefore, care in performing adequate stunning before causing the death of fish is necessary. Many fish-slaughtering methods are employed without involving the stunning process. The conditions under which pre-slaughter and fish-slaughtering operations are carried out, in general, are still precarious [12]. Compromising the final product quality may related to unpreparedness in handling production, infrastructure problems, a lack of storage capacity and inadequate transport to slaughterhouses. These factors can lead to great waste and low yields [13]. At the time of slaughter, the biochemical reactions caused by stress can cause the depletion of muscle glycogen, leading to the rapid onset of rigor mortis and consequently an inadequate pH value in the muscle [14]. As there is no specific law for when fish are to be slaughtered, the choice of method is based on ease of application and reduced cost. However, the method employed must contribute to the final product’s quality.

Given these assumptions, this study aimed to evaluate the influence of different pre-slaughter stunning methods on the meat quality of *A. gigas* from fish farming.

## 2. Materials and Methods

### 2.1. Animal Ethical Statement

The study was conducted by the Universidade Federal de Rondônia (UNIR), and the analyses were conducted by the Laboratório de Análises Físico-Químicas e Microbiológicas (LAFQM/UNIR), Presidente Médici city, RO, Brazil. This research was approved by the Committee for Ethics in the Use of Animals (CEUA/UNIR), with the protocol number 0015/20121. In addition, this study was reported following ARRIVE guidelines.

### 2.2. Pre-Slaughter Stunning Methods

*A. gigas* from a fish farm in the municipality of Porto Velho, RO, Brazil, all from the same excavated tank, were used. For each pre-slaughter method, a total of 12 fish were used, totalling 48 animals with a length of 78 ± 12 cm in the ideal category for commercialisation, 11.1 to 14.0 kg [15,16]. To carry out the experimental procedures, the *A. gigas* were fished using a trawl and taken to the slaughtering site in polyethylene boxes with water. The *A. gigas* had been cultivated for 14 months and fed commercial feed containing 40% crude protein with a granulometry of 2 to 2.5 mm for the initial transition phase (juvenile).

No chemical agent was applied for euthanasia. Only the fishing industry’s conventional methods of stunning and slaughter, in addition to the innovative method of electronarcosis followed by bleeding, were used. The pre-slaughter stunning methods were ice asphyxiation (IA), in which the fish were removed from the box with water and placed in a thermal box in an ice/fish/ice arrangement until death occurred due to a lack of oxygen, as described by Robb et al. [17], Freire and Gonçalves [18], and Angelakopoulos et al. [19]; slaughter via asphyxiation in the air (AA), in which the fish, after being removed from the transport box, were deposited in an empty container where they remained until death; and hypothermia, for which the methodology described by Ashley [20] was adopted, a process in which the fish is submerged in a polyethylene box containing a solution of ice and water (1:1) at a temperature of around 1 °C until death. These animals were subjected to the bleeding process (HS) by cutting the branchial arches.

Regarding the new method of electronarcosis (EE), one container containing approximately 200 L of water and 500 g of sodium chloride NaCl to optimise electrical conduction was used. The fish were immersed in the solution and subjected to an electrical discharge established in preliminary tests: 220 V, 3.5 A, a frequency of 1200 Hz, and an exposure time of 20 s. After the observation of complete numbness (an absence of movement and a loss of balance), the animals were immediately sacrificed by cutting the gill arches for bleeding.

After observing complete desensitisation (movement absence and balance loss) [2], the animals were immediately sacrificed by cutting the branchial arches for bleeding. After slaughter, the *A. gigas* were weighed on a portable digital scale, measured using a millimetre ichthyometer, and each animal was identified with numerical labels as follows: treatment 1 (ice asphyxiation (IA)), treatment 2 (air asphyxiation (AA)), treatment 3 (electronarcosis (EE)) and treatment 4 (hyperthermia followed by bleeding (HB)).

A total of 6 fish from each treatment were used and sent for sample processing; the deboned *A. gigas* were stored in plastic bags with a zip lock closure, receiving the same identification as the animal of origin. After obtaining the first data, still in the vicinity of the fish farm visited, all samples were sent on ice to LAFQM/UNIR, where they were stored in a freezer at 0 °C throughout the experiment.

### 2.3. Data Collection

*A. gigas* were submitted to pH, rigor mortis index (RI %), water activity (Aw), instrumental texture and blood glucose analyses and instrumental colourimetry. The pH measurements started at time zero (immediately after pre-slaughter) and were repeated at 1:00 and 1:30 h at the pre-slaughter site. Then, the samples were sent to LAFQM/UNIR and the analyses continued, being performed at 2:30, 4:30 and 6:30 h. After these times, pH analyses were carried out at 24:00, 36:00, 48:00, 72:00, 96:00, 120:00, 160:00 and 360:00 h after pre-slaughter. Measurements were taken on the right side below the dorsal fin, using a portable pulsation pHmeter (SP1400 model, Sensoglass, São Paulo, Brazil). At each measurement, a new perforation was performed to insert the probe at a different location, at a minimum distance of 1 cm from one to the other to avoid masked data due to possible contamination of the site.

The determination of the RI (%) (Equation (1)) started on the property, at time zero (immediately after slaughter) and followed the exact schedules established for the pH to monitor the animals’ muscular behaviour. This measurement was performed according to the methodology developed by Komolka et al. [21].
RI (%) = (Do − D) × 100/Do(1)
where Do = the distance value that separates the tail base fin from the reference point (time zero); D = the distance separating the tail base fin from the reference point in the selected time intervals.

Then, the Aw of the fillets was determined in (%), and the instrumental texture of the fillets was analysed, always on the right side. Six samples were taken for each treatment, totalling twenty-four samples for the texture profile up to 360:00 h after slaughter. The methodology to determine the texture profile was performed according to Bourne [22]. The attributes compression strength (KgF mm^−2^), firmness (N) and adhesiveness (mm^2^) were evaluated using a TAXT.plus texturometer, using Exponent Stable Micro Systems software (Stable Micro System Ltd., Vienna, Austria) [23]. The samples were set up horizontally on a platform, using a cylindrical probe with a flat end which was ½ inch in diameter. Concerning the conditions of the instrumental texture tests, the pre-test speed was 2 mm s^−1^ and the post-test was 10 mm s^−1^, with a distance of 4.0 mm and a compression force measurement [24]. Measurements started 24 h after slaughter and were repeated up to 360 h after pre-slaughter.

For the blood glucose analysis, blood samples were collected from pre-slaughter fish in which the gills were cut (EE and HB) at the time of bleeding in *A. gigas* slaughtered via asphyxiation methods (IA and AA); blood collection was performed by pulsing the vein located in the tail fin as soon as it was verified using a 5 mL disposable syringe. An Accu-Chek An active glucose meter kit was used, with the value expressed in mg dL^−1^.

Instrumental colourimetry was measured using a portable colourimeter (MINOLTA CR-10, Minolta Camera Co, Osaka, Japan), previously calibrated with a black and white standard before each analysis, operating with a D65 light source, an observation angle of 10° and a 30 mm cell opening. The colour was expressed using the colour standards of the CIELAB system, Comission Internationale de L’Eclairage: L* (brightness, which evaluates a range from 0, which is considered black, to 100, which means white), a* (red-green colour intensity) and b* (yellow-blue colour intensity) [25].

### 2.4. Statistical Analysis

In a completely randomised design, data on pH, RI (%), instrumental texture profile, glucose and colourimetry were preliminarily submitted for an analysis of variance. Then, in the case of normality and significance, Tukey’s test was applied (α = 0.05). For instrumental texture profile delineation, the average (μ) and standard deviation ± SD (σ) of the three repetitions of each fillet at the sampling moment were calculated. Additionally, for the texture profile, to help in interpreting the results for the variables compression strength and adhesiveness, a dispersion analysis was applied.

All statistical analyses were performed using RStudio Development Core Team, version 3.5.3.

## 3. Results

No statistical differences were found between the treatments; the pH varied between (*p* > 0.05) at 120:00 and 168:00, while differences (*p* < 0.05) were found at the other times, especially between the AG and AA treatments. When comparing the average pH over the hours, all treatments (pre-slaughter stunning methods) showed differences (*p* < 0.05). The EE and HB treatments showed more significant reductions at the end of the experimental test, expressing significant reductions at 72:00 (Table 1). It is possible to observe that in all methods, the pH variation started similarly at 0:00 (~6.5) except for the AA fillets. It was found that there was a greater variation in the HB fillets. However, the IA treatment caused a more significant increase in pH at the end of the experimental test (>6.70) at 360:00.

Regarding the RI (%) values, when comparing the averages between treatments, statistical differences were always found (*p* < 0.05). It is worth noting that at 1:00, the IA and HB treatment fillets had higher averages compared to the others; at 72:00, the EE treatment had the lowest average compared to the others and at 120:00, the HB treatment had the highest average compared to the other methods. From 96:00 onwards, the IA treatment tended to reduce the RI (%), while the other AA, EE and HB methods tended to increase the RI (%). Finally, from 96:00 onwards, the RI in the IA method tended to decrease, while the other methods tended to increase the RI (Table 2).

It is possible to observe the tendency of a normal sine wave in the RI (%) for all treatments. The HB treatment from the beginning at 0:00 until the end of the experimental trial at 360:00 showed the highest RI (%), in addition to offering the greatest RI (%) variation compared to the other treatments. However, it is essential to point out that the IA treatment had the lowest RI (%), from the beginning to the end of the experimental trial and the lowest RI (%) variation compared to the other treatments.

Regarding the average Aw (%) variations, when comparing the averages between treatments, there was no statistical difference (*p* > 0.05), and there were no statistical differences (*p* > 0.05) in the average Aw (%) at different times. However, it is possible to mathematically observe a tendency towards a reduction in the Aw in the EE and HB treatments in contrast to the averages of the IA and AA methods that varied little over the hours (Table 3).

Concerning the instrumental texture profile in *A. gigas* fillets, the variables compression strength (KgF mm^−2^), firmness (N) and adhesiveness (mm^2^) were studied. Firstly, when comparing the averages of the treatments, there was no statistical difference in compression strength (*p* > 0.05). However, for firmness, the IA treatment had the highest average at 2528.45 N, and for adhesiveness, the IA and AA treatments had the same values, with higher averages of −60.78 and −59.46 mm^2^ compared to the other slaughter methods. The averages of the attributes differ significantly (*p* < 0.05) as a function of the storage time in the *A. gigas* fillets, so much so that when comparing the averages of the texture attributes over the hours, the treatments expressed growth tending to a sine wave except for HB. However, they expressed growth resembling the IA sine waveform, the AA waveform except for adhesiveness and the EE waveform except for firmness. Therefore, except for HB, the other treatments showed reductions in the averages of compression strength, firmness and adhesiveness (Table 4).

In Figure 1a, it can be observed that the EE and HB treatments resulted in the most adequate compression strength sine waves. The AA treatment resulted in the smallest variation in compression strength compared to the other pre-slaughter stunning methods during 360 h of storage. As compression strength and firmness are matched, the interpretation is similar. In Figure 1b, it is possible to observe that in the HB treatment fillets, the adhesiveness varied little; however, the EE treatment expressed the closest proximity to a sine wave, which showed the greatest variation in adhesiveness during 360 h of storage. The EE adhesiveness started and ended expressing the highest averages compared to the other pre-slaughter stunning methods.

When comparing the blood glucose average, the fish submitted to the HB method expressed a higher average, 104.14 mg dL^−1^, in relation to the other methods (*p* < 0.05), while the fish submitted to the EE method expressed a lower average of 66.33 mg dL^−1^ in relation to the other pre-slaughter stunning methods (Table 5).

Regarding instrumental colourimetry, the parameters of luminosity (L*), chromaticity (a*) (red-green component) and chromaticity (b*) (yellow-blue component) were studied. The EE and HB treatments expressed the highest average L* values (72.05 and 71.19, respectively), although the AA treatment expressed the highest average a* (4.29), while the EE and HB treatments expressed the highest average b* (−12.45 and −10.40, respectively) compared to the other treatments. Furthermore, there was an interaction effect between slaughter method and *A. gigas* fillets (*p* < 0.001) (Table 6). That is, in the *A. gigas* submitted to EE and HB, the fillets tended to be less red. Given this, these methods resulted in less water exposure and the destruction of blood cells, allowing for better luminosity and the preservation of the fillets. It was confirmed that when using the EE method, the blood was drained better, which promotes fillet integrity.

## 4. Discussion

After slaughter, that is, after death, the *A. gigas* organism, like animals, enters an anoxic condition, at which time reactions of the anaerobic glycolysis pathway are triggered in the muscle which lead to the formation of lactic acid, causing a drop in muscle pH. In the current study, the muscle pH values followed the RIISPOA [26], that is, they were below 7.0 throughout the experiment for all treatments (pre-slaughter stunning methods). However, the pH drop is not uniform between different muscles of the same animal, between individuals of the same breed or between those of other breeds and species [27]. The current study had no pH values below 6.0 (Table 1). Therefore, Daskalova [28] ensures that the muscle pH of fish after death will hardly below 6.0, even in full rigor mortis, unlike the meat of mammals. Iguarán et al. [29] found pH averages of 5.89 in *Oreochromis niloticus* fillets which appeared disaggregated, a typical defect of meat from animals that suffered intense stress before slaughter.

Viana et al. [30] reported the occurrence of muscle acidification in *Brycon cephalus* up to 56 h after pre-slaughter stunning with heat shock, 24:30 h for electronarcosis and 26:30 h for CO_2_ asphyxiation. In contrast to what was observed, from those times onwards, there was a gradual increase in pH values for the fish in the EE, AA and HB treatments (Table 1). Viana et al. [30] also reported an increase in muscle pH in *B. cephalus* after the sixth day, as was the case in the current study, in which the rise occurred on the seventh day (Table 1).

The increase in the pH averages in the *A. gigas* fillets may have occurred due to the synthesis of volatile nitrogenous bases (VNBs), such as trimethylamine and ammonia, resulting from the microbial degradation process involving aerobic and facultative anaerobic bacteria [13]. Trimethylamine oxide is a natural component of fish muscle and in the postmortem period, it is reduced to trimethylamine, and ammonia, in turn, results from the degradation of amino acids from muscle proteins [31]. Both compounds interfere with the odour of stored fish. Some authors have questioned the use of pH as a parameter for analysing the meat quality of fish. pH values vary according to the species and capture methods be used and should not be used as the only freshness index method, although it is known that the variation of this parameter during storage may indicate the occurrence of some alteration (biochemical or microbiological) [32].

Average muscle pH variations in *Colossoma macropomum* fillets did not provide a better index for assessing quality since no difference was observed in values in animals that underwent harvesting, 4 h of transport, 24 and 48 h of recovery and slaughter via asphyxiation with CO_2_ and hyperthermia [31]. After pre-slaughter stunning, another significant change in the muscle is the event of rigor mortis, characterised by an extreme and irreversible contraction of the muscle due to the depletion of energy sources when the pH reaches a minimum value. The longer the muscle remains in pre-rigor stage, when there is still muscle flexibility, the longer its shelf life should be [33]. This is because after the period of stiffness, due to pH decay, there is an activation of proteolytic systems, thus causing relaxation and the recovery of muscle elasticity. From then on, the autolysis process takes shape, followed by the action of microbial proteases [34].

Stress before death causes early entry into rigor mortis [31]; paying attention to Figure 1, it can be seen that the IA and AA fish entered full rigor mortis at 1:30 h after pre-slaughter. However, the IA group showed a propensity to enter full rigor mortis before the others; at time zero, 80% of their samples were already in this condition, and 2:30 h after pre-slaughter, 83% of them were already in the post-rigor process, showing relaxation. It is also observed that the animals slaughtered via the EE method showed a 100% accuracy index in all samples 2:30 h after pre-slaughter and remained in it until 36:00 h. Animals pre-slaughtered via HB at 0:00 showed absolute rigor mortis in 50% of the samples, and the remaining animals in the group were close to entering rigor mortis 1:00 h after pre-slaughter stunning. In general, the results for EE point to a more remarkable preservation of muscle glycogen; perhaps because of this, the group took longer to enter rigor mortis [35].

Contrary to what was observed, Digre et al. [36], in a study with Atlantic cod (*Gadus morhua*), found that electronarcosis accelerated the process of entry into rigor mortis when treated with an anaesthetic. Likewise, Morzel et al. [37], with *Psetta maxima*, noted that the electronarcosis group entered rigor mortis before the groups that were slaughtered via cranial percussion and heat shock. There was a rapid reduction in the rigor mortis index in IA, AA and HB fish from 36:00 after pre-slaughter stunning, and EE this reduction from the same time onwards was smoother than in the others (Figure 1).

Mendes et al. [38] listed some causes associated with the speed of pH drop the onset and duration of rigor mortis: handling before slaughter; stress; resistance or an animal’s susceptibility itself to stress; and postmortem temperature. Mendes et al. [38] also mentioned factors such as the body size of the fish, the body fat level and even the species. Finally, Mendes et al. [31], following this line of thought, argued that the entry into and exit from rigor mortis seems to depend on the intensity and duration of pre-slaughter stunning stress.

In studies by Daskalova and Pavlov [39] with *Cyprinus carpio* L. in Bulgaria and Daskalova [28] in a systematic review of several fish species from Europe to South America, the authors pointed out that pre-slaughtering and stunning methods significantly influenced both the texture and the water activity (Aw) of the fillets tested during the days of storage under refrigeration. The texture profile analysis is widely used to evaluate the instrumental texture of fish fillets. However, different configurations and conditions are limiting factors for its use as an official quality-monitoring method. Bernardo et al. [12] studied the instrumental texture of *Oreochromis niloticus* fillets stored at 4.0 °C for 8 days. Different settings were employed, including a compression ratio (CR) between 43.2 and 76.8%, a hold time (HT) between 0.1 and 10 s and a test speed (TS) between 0.3 and 3.7 mm s^−1^, applying the rotating central composite. Pre-slaughter stunning methods influenced the attributions of compression strength, chewiness, adhesiveness and firmness, especially after five days of storage.

Mendes et al. [31] conducted a study examining the pre-slaughter stunning of *Colossoma macropomum* via asphyxiation with CO_2_ and hyperthermia, followed by storage on ice. As in the current study, there was a significant difference in fillet texture, measured using the shear force parameter, between the pre-slaughter methods used. Bahuaud et al. [40] state that stressful and time-consuming pre-slaughter methods resulted in a smoother (softened) muscle texture and a shorter shelf life in a study carried out with salmonids. Cassol et al. [41] verified the effects of different stunning and pre-slaughter methods on the meat quality of Amazonian guinea fowl, female *Pseudoplatystoma fasciatum* × male *Leiarius marmoratus*, stored on ice for 18 days. A total of 90 specimens were divided into three groups and subjected to stunning via CO_2_-saturated water, hyperthermia on ice or asphyxiation in air. The methods significantly influenced the meat quality, especially those with bleeding reinforcement. Fish stunned on ice showed better results than asphyxiation with water saturated with CO_2_ [41].

Consumers rigorously assess the quality of fish from fishing and aquaculture every day, mainly considering freshness, appearance and texture. The application of stunning methods is an essential factor in fish quality because it maintains crucial organoleptic characteristics that will influence commercialisation. Given this, Presenza et al. [42] compared some pre-slaughter stunning methods, including eugenol stunning and hyperthermia. The authors found that slaughter preceded by the application of Eugenol resulted in full rigor mortis after 6 h, and when using the hyperthermia method, it lasted 4 h. The pre-slaughter stunning method augmented with eugenol caused less stress, positively influencing the duration of rigor mortis stages.

Regarding instrumental colourimetry, in a study by Bassil et al. [43], *A. gigas* were subjected to the bleeding method without first undergoing pre-slaughtering. There was no difference in brightness on 0, 3, 6 and 9 days. The observed similarities may be related to residual blood in the carcass. Residual blood and the consequent presence of haem pigments can contribute to a darker meat surface and similar luminosity values. According to Erikson et al. [44], who evaluated the effect of tapping on Atlantic salmon (*Salmo salar*), they did not observe significant differences in luminosity in fillets from bled and non-bled animals. The same authors reported higher luminosity values in Atlantic salmon pre-slaughtered via hyperthermia followed by bleeding. Sternisa et al. [45] found similar luminosity values in carp (*Cyprinus carpio*) pre-slaughtered under hyperthermia followed by bleeding compared to non-bled fish during 12 days of refrigerated storage (4 °C). In contrast, Addeen et al. [46] reported a decrease in lightness in bled and non-bled areas in several animal species.

Concerning red colouration, *A. gigas* exhibited more significant redness following pre-slaughter stunning methods applying bleeding and more yellowish without bleeding [43]. This result can be attributed to the maintenance of haem pigments such as haemoglobin and myoglobin in the muscle, contributing to an increase in red colouration. Sternisa et al. [45] observed that non-bled fish, carp (*Cyprinus carpio*), exhibited higher a* values than bled carp on the third day of refrigerated storage. Jiang et al. [47] studied the effect of blood deposition on the pulp quality of yellowtail Seriola quinqueradiata. They reported a more significant redness in a non-bled yellowtail muscle on the third day of storage (4 °C). In contrast, Erikson et al. [44] evaluated the effect of tapping on fillets of Atlantic salmon and reported a* values in unbled and bled samples.

Muscle pH can influence colour by influencing oxygen consumption and metmyoglobin-reducing activity [48]. pH values close to neutrality increase the ability of mitochondria to compete with myoglobin for available oxygen, resulting in more metmyoglobin formation [49]. Baldi et al. [50] evaluated the influence of pre-slaughter stunning methods on animal welfare aspects and on the physical and chemical parameters of frozen *Rachycentron canadum* during frozen storage. The methods applied were electronarcosis, hyperthermia and CO_2_ asphyxia after bleeding performed by cutting the gills and storing the fish at −18 °C for 180 days. During pre-slaughter stunning, the water quality and time to reach clinical indicators of unconsciousness were observed. During the storage period, every 60 days, samples were taken, and the pH, volatile total basic nitrogen (VTBN), thiobarbituric acid reactive substances (TBARS), protein (actin and myosin) denaturation, fatty acid composition, colour (L*, a* and b*), texture, dripping loss (DL), cooking loss (CL) and water holding capacity (WHC) were analysed. The parameters VTBN, TBARS, DL, texture, actin, myosin, pH, L*, a* and b* were not affected by the stunning methods, while the CL was significantly higher in the fillets of fish stunned via electronarcosis, not altering the shelf life of the fillets. Based on the behaviour of the parameters analysed by Melo et al. [50], electronarcosis seemed to be a suitable stunning method for *R. canadum* because it promoted rapid unconsciousness and did not compromise meat quality.

Regarding yellow colour, the b* values were higher in fish that underwent methods of slaughter under stunning without bleeding, such as electronarcosis. Similar to redness, higher b* values can be attributed to the presence of haem pigments (haemoglobin and myoglobin) in the samples. There is a high content of polyunsaturated fatty acids in *A. gigas* fillets [43]. This, in turn, may favour lipid-oxidation-induced myoglobin oxidation [51] and metmyoglobin accumulation in fillets [52], contributing to increased b*a values throughout the last 10 days of refrigerated storage.

In partial agreement, Erikson et al. [44] evaluated the effect of tapping in Atlantic salmon and reported similar b* values in unbled and bled samples. On the contrary, Ramos et al. [53] evaluated meat colour and pigment levels in bled and non-bled *Rana catesbeiana* and reported higher b* values in the bled samples. During storage, a decrease in yellowing (b* values) was observed in *A. gigas* fillets, while the b* values remained stable. Yellowing losses can be attributed to lipid oxidation and changes in haem pigments [46], decreasing yellowing. Nguyen and Phan [54] reported b* values in unbled *Rachycentron canadum* fillets compared to bled samples during 24 weeks of storage. Furthermore, Digre et al. [36] reported an increase in b* values in bled and unbled Atlantic cod fillets after seven days of refrigerated storage. Sternisa et al. [45] documented greater yellowing in non-bled carp fillets over 12 days of refrigerated storage.

In view of the colourimetry data discussed, the EE and HB methods exposed less water from the fillets and avoided the destruction of animal organism cells, allowing for greater preservation. In this way, these methods allow for better luminosity. Additionally, for EE, in addition to draining more blood from the fish, which promotes fillet integrity, it provided more juiciness and tenderness to the meat.

## 5. Conclusions

Electronarcosis was an excellent pre-slaughter stunning method; the fish subjected to this method did not suffer stress during slaughter. Furthermore, during the 15 days of storage under refrigeration, the fish fillets subjected to electronarcosis showed better preservation, juiciness, and tenderness.

## Figures and Tables

**Figure 1 animals-14-01155-f001:**
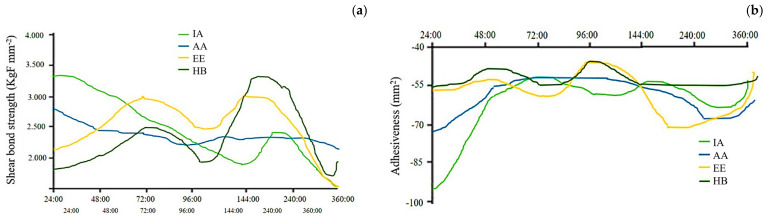
Graphical representation of (**a**) compression variations and (**b**) adhesiveness variations in *Arapaima* gigas fillets under refrigeration for up to 360 h of storage, using different pre-slaughter stunning methods.

**Table 1 animals-14-01155-t001:** Variations in the average muscle pH of *Arapaima gigas* fillets submitted to pre-slaughter stunning by ice asphyxia (IA), air asphyxia (AA), electronarcosis (EE) or hyperthermia followed by bleeding (HB) at different sampling times.

Hours	Treatments				*p*-Value
	IA	AA	EE	HB	
0:00	6.60 ± 0.14 ^aA^	6.49 ± 0.16 ^bAB^	6.59 ± 0.16 ^aA^	6.62 ± 0.14 ^aA^	0.0233
1:00	6.38 ± 0.14 ^bAB^	6.47 ± 0.16 ^aAB^	6.43 ± 0.16 ^aAB^	6.29 ± 0.13 ^bB^	0.0367
1:30	6.22 ± 0.11^bB^	6.41 ± 0.15 ^abAB^	6.55 ± 0.16 ^aAB^	6.34 ± 0.13 ^abB^	0.0299
2:30	6.26 ± 0.13 ^bAB^	6.31 ± 0.15 ^bB^	6.37 ± 0.16 ^abAB^	6.56 ± 0.14 ^aAB^	0.0370
4:30	6.33 ± 0.14 ^ab^	6.26 ± 0.15 ^bB^	6.13 ± 0.15 ^bB^	6.43 ± 0.13 ^aAB^	0.0214
6:30	6.16 ± 0.13 ^bB^	6.50 ± 0.16 ^aAB^	6.49 ± 0.16 ^aAB^	6.40 ± 0.13 ^abAB^	0.0288
24:00	6.37 ± 0.14 ^bAB^	6.38 ± 0.15 ^bAB^	6.45 ± 0.16 ^abAB^	6.63 ± 0.14 ^aA^	0.0202
36:00	6.30 ± 0.13 ^bAB^	6.40 ± 0.15 ^abAB^	6.37 ± 0.16 ^abB^	6.42 ± 0.13 ^aAB^	0.0150
48:00	6.33 ± 0.14 ^bAB^	6.58 ± 0.16 ^aA^	6.41 ± 0.16 ^bAB^	6.39 ± 0.13 ^bB^	0.0226
72:00	6.34 ± 0.13 ^aAB^	6.35 ± 0.15 ^aAB^	6.31 ± 0.16 ^aB^	6.30 ± 0.13 ^aB^	0.0221
96:00	6.21 ± 0.13 ^bAB^	6.41 ± 0.15 ^aAB^	6.37 ± 0.16 ^abB^	6.35 ± 0.13 ^abB^	0.0249
120:00	6.28 ± 0.13 ^aAB^	6.27 ± 0.15 ^aB^	6.28 ± 0.15 ^aB^	6.33 ± 0.13 ^aB^	0.0266
168:00	6.29 ± 0.13 ^aAB^	6.36 ± 0.15 ^aAB^	6.31 ± 0.16 ^aB^	6.30 ± 0.13 ^aB^	0.0148
360:00	6.72 ± 0.14 ^aA^	6.43 ± 0.15 ^abAB^	6.44 ± 0.16 ^abAB^	6.32 ± 0.13 ^bB^	0.0332
*p* value	0.0289	0.02914	0.0303	0.0220	-

Averages followed by different letters in same row (a,b) and in the same column (A,B) are different from each other in Tukey’s test (*p* < 0.05).

**Table 2 animals-14-01155-t002:** Variations in the average rigor mortis index (RI) (%) in *Arapaima gigas* fillets over the course of hours for pre-slaughter stunning via ice asphyxia (IA), air asphyxia (AA), electronarcosis (EE) and hyperthermia followed by bleeding (HB) at different sampling times.

Hours	Treatments				*p* Value
	IA	AA	EE	HB	
0:00	3.30 ± 0.86 ^bB^	9.33 ± 2.52 ^aA^	8.58 ± 2.40 ^aA^	6.17 ± 1.23 ^a bA^	0.0437
1:00	1.33 ± 0.34 ^bC^	5.83 ± 1.57 ^aA^	6.92 ± 1.94 ^aA^	0.67 ± 0.13 ^bB^	0.0404
1:30	0.00 ± 0.00 ^bC^	0.00 ± 0.00 ^bB^	2.50 ± 0.70 ^aAB^	0.17 ± 0.03 ^bB^	0.0420
2:30	1.67 ± 0.43 ^aC^	0.00 ± 0.00 ^bB^	0.00 ± 0.00 ^bB^	0.50 ± 0.10 ^bB^	0.0444
4:30	1.83 ± 0.48 ^aC^	0.00 ± 0.00 ^bB^	0.00 ± 0.00 ^bB^	0.58 ± 0.12 ^bB^	0.0404
6:30	1.25 ± 0.33 ^aC^	0.00 ± 0.00 ^bB^	0.00 ± 0.00 ^bB^	0.08 ± 0.02 ^bB^	0.0400
24:00	0.92 ± 0.24 ^bC^	0.00 ± 0.00 ^bB^	0.50 ± 0.00 ^bB^	4.18 ± 0.84 ^aA^	0.0410
36:00	2.50 ± 0.65 ^a bB^	0.00 ± 0.00 ^bB^	0.00 ± 0.00 ^bB^	5.33 ± 1.07 ^aA^	0.0425
48:00	1.08 ± 0.28 ^bC^	7.18 ± 1.94 ^aA^	2.78 ± 0.78 ^abAB^	5.75 ± 1.15 ^aA^	0.0255
72:00	5.25 ± 1.37 ^aAB^	5.17 ± 1.40 ^aA^	3.75 ± 1.05 ^bAB^	6.00 ± 1.20 ^aA^	0.0290
96:00	5.08 ± 1.32 ^bAB^	8.67 ± 2.34 ^aA^	6.92 ± 1.94 ^bA^	5.50 ± 1.10 ^bA^	0.0260
120:00	6.33 ± 1.64 ^aAB^	6.75 ± 1.82 ^aA^	6.92 ± 1.94 ^aA^	5.90 ± 1.18 ^bA^	0.0195
168:00	5.92 ± 1.54 ^aAB^	6.00 ± 1.62 ^aA^	5.67 ± 1.59 ^aA^	6.33 ± 1.27 ^aA^	0.0199
360:00	4.17 ± 1.08 ^cB^	9.08 ± 2.45 ^aA^	6.92 ± 1.94 ^bA^	6.77 ± 1.35 ^bA^	0.0305
*p* value	0.0355	0.0440	0.0406	0.0480	-

Averages followed by different letters in the same row (a,b,c) and in the same column (A,B,C) are different from each other in Tukey’s test (*p* < 0.05).

**Table 3 animals-14-01155-t003:** Variations in average water activity (Aw) (%) in *Arapaima gigas* fillets over the course of hours for pre-slaughter stunning via ice asphyxia (IA), air asphyxia (AA), electronarcosis (EE) and hyperthermia followed by bleeding (HB) at different sampling times.

Hours	Treatments				*p* Value
	IA	AA	EE	HB	
72:00	98.10 ± 5.89 ^aA^	98.00 ± 5.49 ^aA^	97.77 ± 5.87 ^aA^	95.80 ± 6.42 ^aA^	>0.05 ^ns^
96:00	97.66 ± 5.86 ^aA^	97.10 ± 5.44 ^aA^	95.55 ± 5.73 ^aA^	96.00 ± 6.43 ^aA^	>0.05 ^ns^
120:00	97.00 ± 5.82 ^aA^	96.00 ± 5.38 ^aA^	96.10 ± 5.77 ^aA^	96.30 ± 6.45 ^aA^	>0.05 ^ns^
144:00	96.33 ± 5.78 ^aA^	95.50 ± 5.35 ^aA^	94.60 ± 5.68 ^aA^	94.40 ± 6.32 ^aA^	>0.05 ^ns^
360:00	95.50 ± 5.73 ^aA^	95.10 ± 5.32 ^aA^	92.80 ± 5.57 ^aA^	93.00 ± 6.23 ^aA^	>0.05 ^ns^
*p* value	>0.05 ^ns^	>0.05 ^ns^	>0.05 ^ns^	>0.05 ^ns^	-

Averages followed by different letters in the same row (a) and in the same column (A) are different from each other in Tukey’s test (*p* < 0.05). ns = Not significant.

**Table 4 animals-14-01155-t004:** Variations in the averages of instrumental texture profile variables in *Arapaima gigas* fillets under refrigeration for up to 360 h of storage, using different pre-slaughter stunning methods.

Treat. *	Hours	Attributes (ẋ_Day_)		
		Compression Strength (KgF mm^−2^)	Firmness (N)	Adhesiveness (mm^2^)
	24:00	3.349.53 ± 911.74 ^a^	3.548.16 ± 875.33 ^a^	−93.99 ± 14.24 ^a^
	48:00	2.770.36 ± 754.64 ^b^	2.734.58 ± 674.62 ^b^	−61.05 ± 9.25 ^b^
	72:00	2.255.03 ± 613.82 ^b^	2.402.06 ± 592.59 ^b^	−49.79 ± 7.54 ^c^
IA	96:00	2.738.57 ± 745.44 ^b^	2.749.43 ± 678.28 ^b^	−55.22 ± 8.37 ^c^
	144:00	2.000.98 ± 544.67 ^b^	2.139.97 ± 527.93 ^b^	−50.74 ± 7.69 ^c^
	240:00	2.375.86 ± 646.71 ^b^	2.504.72 ± 617.91 ^b^	−61.53 ± 9.32 ^b^
	360:00	1.620.23 ± 441.03 ^c^	1.620.23 ± 399.70 ^c^	−53.15 ± 8.05 ^c^
ẋ_Treat._		2.444.37 ± 665.36 ^A^	2.528.45 ± 623.77 ^A^	−60.78 ± 9.21 ^A^
*p* value_Period_		0.0225	0.0420	0.0175
	24:00	2.758.15 ± 438.27 ^a^	2.884.12 ± 445.31 ^a^	−73.66 ± 11.60 ^a^
	48:00	2.380.75 ± 378.30 ^b^	2.437.75 ± 376.39 ^b^	−55.47 ± 8.31 ^b^
	72:00	2.562.05 ± 407.11 ^ab^	2.501.86 ± 386.29 ^b^	−55.86 ± 8.37 ^b^
AA	96:00	2.528.42 ± 401.76 ^b^	2.654.66 ± 409.88 ^ab^	−54.13 ± 8.11 ^b^
	144:00	2.452.66 ± 389.73 ^b^	2.452.66 ± 378.69 ^b^	−56.16 ± 8.42 ^b^
	240:00	2.548.83 ± 405.01 ^b^	2.605.80 ± 402.33 ^ab^	−65.60 ± 9.83 ^ab^
	360:00	1.586.84 ± 252.63 ^c^	1.620.39 ± 250.19 ^c^	−55.34 ± 8.30 ^b^
ẋ_Treat._		2.451.03 ± 389.47 ^A^	2.402.53 ± 370.95 ^B^	−59.46 ± 8.91 ^A^
*p* value_Period_		0.0131	0.0107	0.0464
	24:00	2.229.21 ± 529.88 ^b^	2.440.04 ± 507.53 ^b^	−54.60 ± 12.12 ^b^
	48:00	2.223.81 ± 258.60 ^b^	2.239.36 ± 465.79 ^b^	−48.68 ± 10.80 ^c^
	72:00	2.865.11 ± 681.04 ^a^	2.719.74 ± 565.71 ^a^	−57.78 ± 12.84 ^b^
EE	96:00	2.510.10 ± 596.65 ^ab^	2.479.53 ± 515.74 ^b^	−48.80 ± 10.83 ^c^
	144:00	2.195.03 ± 521.76 ^bc^	2.219.90 ± 461.74 ^b^	−49.29 ± 10.94 ^c^
	240:00	2.632.13 ± 625.66 ^ab^	2.796.24 ± 581.62 ^a^	−65.10 ± 14.45 ^a^
	360:00	1.716.31 ± 407.97 ^c^	1.729.89 ± 567.81 ^c^	−59.60 ± 13.23 ^b^
ẋ_Treat._		2.374.96 ± 564.53 ^A^	2.338.81 ± 486.47 ^B^	−54.84 ± 12.17 ^B^
*p* value_Period_		0.0149	0.0182	0.0440
	24:00	1.796.53 ± 453.80 ^c^	1.993.76 ± 382.80 ^c^	56.71 ± 5.03 ^a^
	48:00	2.289.10 ± 578.23 ^bc^	2.266.55 ± 435.18 ^bc^	−50.33 ± 4.47 ^b^
	72:00	2.555.20 ± 645.44 ^b^	2.511.67 ± 482.24 ^b^	−56.00 ± 4.97 ^a^
HB	96:00	1.949.77 ± 492.51 ^bc^	2.585.71 ± 496.46 ^b^	−45.68 ± 4.05 ^c^
	144:00	3.122.74 ± 788.80 ^a^	3.122.74 ± 599.56 ^a^	−50.88 ± 4.52 ^b^
	240:00	3.303.86 ± 834.55 ^a^	3.414. 22 ± 655.53 ^a^	−55.95 ± 5.00 ^a^
	360:00	2.169.31 ± 547.97 ^bc^	2.114.13 ± 405.91 ^bc^	−55.46 ± 4.92 ^a^
ẋ_Treat._		2.432.43 ± 614.43 ^A^	2.455.22 ± 471.40 ^AB^	−53.00 ± 4.71 ^B^
*p* value_Period_		0.0150	0.0167	0.0409

* Pre-slaughter stunning methods: ice asphyxiation (IA), air asphyxiation (AA), electronarcosis (EE) and hyperthermia followed by bleeding (HB); averages followed by different letters between periods (a,b,c) and in average treatments (A,B) are different from each other in Tukey’s test (*p* < 0.05); ẋDays = average of the days.

**Table 5 animals-14-01155-t005:** Variations in average blood glucose in *Arapaima gigas* submitted to different pre-slaughter stunning methods; values expressed in mg dL^−1^.

Hour	Treatments *				*p* Value
	IA	AA	EE	HB	
0:00	75.00 ± 8.55 ^b^	77.97 ± 8.89 ^b^	66.33 ± 7.56 ^c^	104.14 ± 11.90 ^a^	0.0740

* Pre-slaughter stunning methods: ice asphyxiation (IA), air asphyxiation (AA), electronarcosis (EE) and hyperthermia followed by bleeding (HB); averages followed by different letters between periods (a,b,c) are different from each other in Tukey’s test (*p* < 0.05).

**Table 6 animals-14-01155-t006:** Instrumental colourimetry in *Arapaima gigas* fillets at 360 h under refrigeration for up to 360 h of storage, submitted to different pre-slaughter stunning methods.

Variables	Treatments *				*p* Value
	IA	AA	EE	HB	
L*	63.72 ± 2.67 ^b^	59.57 ± 2.49 ^b^	72.05 ± 3.01 ^a^	71.19 ± 2.98 ^a^	0.0325
a*	3.19 ± 0.25 ^b^	4.29 ± 0.33 ^a^	3.23 ± 0.25 ^b^	3.08 ± 0.24 ^b^	0.0290
b*	−5.33 ± 0.20 ^b^	−4.45 ± 0.16 ^b^	−12.45 ± 0.46 ^a^	−10.40 ± 0.38 ^a^	0.0267
Probabilities					
Pre-slaughter stunning methods (T)	<0.0001	<0.0001	<0.0001	<0.0001	-
Colourimetry (C)	<0.0001	<0.0001	<0.0001	<0.0001	-
T × C interaction	<0.0001	<0.0001	<0.0001	<0.0001	-

* Pre-slaughter stunning methods: ice asphyxiation (IA), air asphyxiation (AA), electronarcosis (EE) and hyperthermia followed by bleeding (HB); averages followed by different letters between periods (a,b) are different from each other in Tukey’s test (*p* < 0.05).

## Data Availability

Data are contained within the article.

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
