# Peer review of "Pre-Slaughter Stunning Methods Influence the Meat Quality of *Arapaima gigas* Fillets"

_animals, 2024, doi:10.3390/ani14081155_

Round 1

Reviewer 1 Report

Comments and Suggestions for Authors

Pre-slaughter stunning methods influence the meat quality of Arapaima gigas fillets

General evaluation:

The subject of the manuscript is interesting and relevant to good aquaculture practices. However, some parts of the manuscript should be improved. Furthermore, the manuscript must be improved in the English language.

Specify evaluation:

Abstract:  What is the conclusion of the study?

Introduction: ok

M & M:

-          The nomenclatures do not correspond to the words in English.

-          I didn't understand the treatment 4. In all the treatments after stunning, was there no bleeding performed?

-          Fillets stored in a freezer at 0C? Wouldn't it be a refrigerator?

Results:

The tables and figures are very confusing. It is not possible to clearly verify what the statistical differences are. Show only the figures. Furthermore, the authors did not show whether there was a difference between treatments, time or interaction.

Discussion:

Ok

Conclusion:

Authors should not summarize the results and only conclude the study. There is a lack of a clear conclusion as to what was the best stunning method for the fish species.

Comments on the Quality of English Language

The manuscript must be improved in the English language.

Author Response

Dear Editor, the questions and suggestions for adjustments were answered point-by-point. Please see the attachment.

Additionally, improvements to the original file have been made and marked in green.

Sincerely

Reviewer 2 Report

Comments and Suggestions for Authors

Author Response

(The authors gave the same response as above.)

Reviewer 3 Report

Comments and Suggestions for Authors

This manuscript described the quality changes of Arapaima gigas affected by different stunning methods. This manuscript was written well. However, some details need to be added. Here are the reviewer’s detailed comments. Please consider the reviewer’s opinion in revising this manuscript to make it more comprehensive.

General and specific comments:

1. Introduction: Many papers have reported the effect of different stunning methods on fish quality, including freshwater fish and saltwater fish. Please discuss about them.

2. Section 2.2: What is the length of the fish?

3. Fig.1: Table 1 and figure 1b are duplicated, Table 2 and figure 1a are duplicated, please delete figure 1.

Author Response

(The authors gave the same response as above.)

Reviewer 4 Report

Comments and Suggestions for Authors

Manuscript ID: animals-2693704 – review

Title:  Pre-slaughter stunning methods influence the meat quality of Arapaima gigas fillets

The manuscript concerns the impact of different stunning methods on the meat quality of Arapaima gigas fillets. The Authors used ice asphyxia, air asphyxia, electronarcosis and hypothermia followed by bleeding and determined their impact on changes in pH, rigor mortis index, water activity, instrumental texture, blood glucose and instrumental colorimetry. The undertaken research addresses an important issue because, as the Authors emphasize in the introduction, pre-slaughter processing affects the quality of the obtained raw material. However, it must be noted that A. gigas is a traditional Amazonian dish and is not an important raw material on the international arena. Additionally, the Authors state that the currently environmental authorities, consider the natural populations of A. gigas sensitive to extinction, others consider it an invasive species in some watersheds. Despite the low popularity of this fish, in the opinion of the Reviewer, the results of the undertaken research are very important and can be applied to other fish species. Therefore, the Reviewer recommend above mentioned paper for publication in Animals after minor revision.

Title:

·    according to the Reviewer, a better title would be: “Effect of pre-slaughter stunning methods on meat quality of Arapaima gigas fillets”. This is only a suggestion and the final decision is up to the Authors.

Introduction:

·     the Reviewer suggest moving the sentence about humane stunning and slaughter (“However, stunning methods and slaughter by…and less suffering to the fishes”, lines 46-49) after listing all possible methods (“Fish can subjected to both phases or directly to the second phase”, lines 54-55). The Authors should remove the word “However”, while moving the sentence;

·       it would be good to combine two paragraphs into one: line 50-59 and 60-69;

·      the Authors repeated the frequency factor twice (“frequency (Hertz) [5], frequency (Hertz) [6]”, line 62);

·    in the Reviewer’s opinion, the text on Arapaima gigas (lines 70-78) should be expanded to include the basic meat quality parameters of this fish from the literature.

Materials and Methods:

·     the reviewer suggests that the description of stunning methods lacks details on how to perform electronarcosis (lines 111-122);

·     on line 185 (”for the instrumental texture…”) the sentence begins with a lowercase letter.

General comment

According to the Reviewer, the chapter contains all the necessary information. The Authors correctly presented the sample collection scheme, breeding conditions, stunning methods (except electronarcosis) and slaughter. The chapter also includes a detailed description of analytical and statistical methods.

Results

·       in the sentence on lines 257-258, the Authors wrote that “it was observed that the EE and HS treatments expressed the most adequate compression strength sine waves according to the literature”, but they didn’t cite any item. In the Reviewer’s opinion, the reference text should be provided or the sentence should be reworded.

General comment

The results were presented in the form of comprehensive tables and figures. Despite many numerical data, the description took into account the most important differences between the analyzed factors.

Discussion

General comment

In the Reviewer's opinion, the chapter was prepared very logically. Due to the low availability of literature regarding the quality of A. gigas meat, the Authors quoted the results of studies done on other fish species. A correct analysis of the obtained values of quality parameters in relation to other test results was made.

Conclusions

General comment

The chapter contains the most important conclusions of the research performed. At the same time, the authors summarized the obtained results of laboratory analysis in a good way.

References

General comment

The Authors have cited 54 items of literature that match the content of the manuscript. The Reviewer would like to commend that most of the works cited were from the last 10 years.

General comments on the whole manuscript:

·    words separated by a hyphen appeared throughout the text, e.g. line 133 ex-periment, line 136- instru-mental, line 149 ac-cording, line 178 Internatio-nale, 183 pro-file. The Reviewer suggests checking the entire content of the manuscript in this regard;

·     there should be a check of the degrees of Celcius in the content of the work, because often there was unnecessary emphasis, e.g. line 133 0º C.

These minor errors may only be visible in the file downloaded by the Reviewer, but this should be checked.

Author Response

(The authors gave the same response as above.)

Round 2

Reviewer 3 Report

Comments and Suggestions for Authors Fig. 1: The author said that the "the table presents the statistical differences and the graphs present the dispersion, that is, the variations in the averages over time." However, the table also showed the variations in the averages over time (in column). I still think that fig. 1 is a duplicate.

Author Response

Dear Editor,

We got together and went back and decided to agree with the reviewer 3 that Figure 1 is unnecessary because the data is already explained in Table 1.

Best regards
